# Design of a Compact Analog Complex Correlator for Millimeter-Wave Radiation Temperature Measurement System

**DOI:** 10.3390/mi14040867

**Published:** 2023-04-17

**Authors:** Wangdong He, Anyong Hu, Chen Dong, Xi Chen, Jianhao Gong, Jungang Miao

**Affiliations:** School of Electronics and Information Engineering, Beihang University, Beijing 100191, China; hwd19930315@buaa.edu.cn (W.H.); phoenixdc@buaa.edu.cn (C.D.); chenxi0913@buaa.edu.cn (X.C.); gongjh@buaa.edu.cn (J.G.); jmiaobremen@buaa.edu.cn (J.M.)

**Keywords:** analog complex correlator, six-port, Ka band, noise measurement, microwave radiometer, thermometer

## Abstract

Human body temperature is a fundamental physiological sign that reflects the state of physical health. It is important to achieve high-accuracy detection for non-contact human body temperature measurement. In this article, a Ka band (32 to 36 GHz) analog complex correlator using the integrated six-port chip is proposed, and a millimeter-wave thermometer system based on the designed correlator is completed for human body temperature measurement. The designed correlator utilizes the six-port technique to achieve large bandwidth and high sensitivity, and miniaturization of the correlator is achieved through an integrated six-port chip. By performing the single-frequency test and the broadband noise measurement on the correlator, we can determine that the dynamic range of input power of the correlator is −70 dBm to −35 dBm, and the correlation efficiency and equivalent bandwidth are 92.5% and 3.42 GHz, respectively. Moreover, the output of the correlator varies linearly with the input noise power, which reveals that the designed correlator is suitable for the field of human body temperature measurement. Then, a handheld thermometer system, with a size of 140 mm × 47 mm × 20 mm, is proposed using the designed correlator, and the measurement results show that the temperature sensitivity of the thermometer is less than 0.2 K.

## 1. Introduction

Microwave radiometer has been widely used in astronomical observation [1], atmospheric remote sensing [2], security sensing [3,4], temperature measurement [5], and other scenarios. Since body temperature is one of the vital signs of the human body and an important indicator to measure the condition of the body, the accurate detection of human body temperature can assist doctors to diagnose and treat patients faster. Moreover, accurate temperature measurements for different parts of the human body can assist doctors to further understand the physiological status of patients. In the field of human body temperature measurement, microwave radiation thermometry, as a non-contact rapid detection method of human body temperature, has received a great deal of attention in recent years. Compared with infrared temperature measurement [6], which is the main method in the field of non-contact rapid detection of human body temperature, microwave radiation temperature measurement has obvious advantages in clothing penetration, influence of surrounding environment, temperature measurement distance, and temperature measurement accuracy [7].

At present, the microwave radiation temperature measurement system based on the total power radiometer and Dicke radiometer has achieved a series of achievements [8,9,10]. However, the total power radiometer and Dicke radiometer has large temperature measurement error and low temperature measurement accuracy. The microwave radiation temperature measurement system using a correlation radiometer has higher temperature sensitivity, and the negative effect of gain fluctuation on temperature measurement accuracy is also lower [11,12].

In the correlation radiometer, the correlator is one of the key components by performing cross-correlation measurement on two input signals [13]. According to the different realization methods, correlators can be divided into digital correlators and analog correlators. The digital correlator has the advantages of high stability and flexible configuration, but it will introduce quantization noise in the quantization process, resulting in deterioration in correlation efficiency and temperature sensitivity [14,15]. The analog correlator has the advantages of large bandwidth, high sensitivity, and low cost [16,17,18]; therefore, it is adopted in the correlation radiometer system to meet the high-precision requirements of human body temperature measurement.

In addition, both direct multiplication technology [19,20] and add-and-square technology [21,22] can be used to implement the analog correlator. Due to the need to customize the special multiplier chip, the cost and difficulty of the analog correlator, which uses direct multiplication technology, to achieve broadband is relatively high [23]. By comparison, the analog correlator using add-and-square technology is generally based on the six-port technique [24]; due to its low cost and ease in achieving large bandwidth, the six-port technique is widely used in the fields of reflectometers [25,26,27], receivers [28,29,30], correlators [31,32], and so on. Therefore, the add-and-square analog complex correlator makes it easier to achieve large bandwidth, and the sensitivity of a correlation radiometer is inversely proportional to the root of the bandwidth [21], which also means that the radiometer can achieve higher temperature sensitivity. In summary, the use of an analog complex correlator can improve the working bandwidth of the correlation radiometer system, thus improving the temperature sensitivity of the system. In addition, compared with the total power radiometer, the correlated radiometer based on analog complex correlator has better temperature sensitivity in theory. On the other hand, the correlation radiometer system applied to human body temperature measurement needs to meet the application requirements of handheld, so the volume of the radiometer needs to be miniaturized, which also puts forward the demand for miniaturization of the analog correlator.

In this paper, an add-and-square analog complex correlator is designed using the six-port technique and its operating frequency is 32–36 GHz. The six-port network in the designed correlator is implemented using the integrated six-port chip, which has been introduced in our previous work [33]. The basic theory of the correlator is introduced and the influence of reflected signal on the correlator during mismatch of the detection circuit is analyzed, which provides assistance for the integrated design of the analog correlator. Further, the analysis results show that the reverse isolation between the detection circuit and the six-port network is necessary so that the correlator can achieve better performance. Then, the designed correlator is measured under the point frequency signal and noise signal so the dynamic range of input power, equivalent noise bandwidth, and correlation efficiency of the designed correlator can be obtained. Finally, the correlation radiometer based on the correlator is introduced and processed, and the temperature sensitivity of the radiometer is measured.

The rest of this article is organized as below. In Section 2, we introduce the basic theory of the correlator and then analyze the influence of reflected signal on the correlator during mismatch of the detection circuit to guide the design of the analog correlator. Section 3 presents the design and integration of the designed analog complex correlator. In Section 4, the performance of the proposed correlator is measured and analyzed. In Section 5, a correlation radiometer system based on the designed correlator for human body temperature measurement is introduced, and the temperature sensitivity is tested. Finally, the paper is concluded in Section 6.

## 2. Analysis of the Analog Complex Correlator

Generally, an ideal correlator can be used to measure the amplitude and phase information between two input signals. The typical block diagram of the add-and-square analog complex correlator [34,35] is illustrated in Figure 1, and we can see that the correlator is mainly composed of the six-port network, detection circuit, and differential amplifier circuit.

### 2.1. Theory of the Analog Complex Correlator

In this section, the basic theory of the correlator will be introduced in the ideal case, assuming that two input signals, $S_{1}$ and $S_{2}$, are represented by $V_{1}\left(t\right)$ and $V_{2}\left(t\right)$, respectively. Then, the output of the signal after passing through the six-port network is
(1)$$\begin{array}{l} S_{3}=\frac{1}{2}\left[V_{1}\left(t\right)+{\hat{V}}_{2}\left(t\right)\right]\\ S_{4}=\frac{1}{2}\left[{\hat{V}}_{1}\left(t\right)+V_{2}\left(t\right)\right]\\ S_{5}=\frac{1}{2}\left[{\hat{V}}_{1}\left(t\right)+{\hat{V}}_{2}\left(t\right)\right]\\ S_{6}=\frac{1}{2}\left[-V_{1}\left(t\right)+V_{2}\left(t\right)\right] \end{array}$$
where ${\hat{V}}_{1}\left(t\right)$ means the Hilbert transform of $V_{1}\left(t\right)$. Here, the Hilbert transform is used to represent a signal phase shift of 90 degrees, which is convenient for subsequent derivation in the article. Assuming the signal conversion coefficient is 1, then the useful output direct current (DC) voltage after the square-law detector and low-pass filter (LPF) can be expressed [36] as
(2)$$\begin{array}{l} V_{3}=E\left[v_{3}\left(t\right)\right]=\frac{1}{4}E\left[V_{1}^{2}\left(t\right)\right]+\frac{1}{4}E\left[{\hat{V}}_{2}^{2}\left(t\right)\right]+\frac{1}{2}E\left[V_{1}\left(t\right){\hat{V}}_{2}\left(t\right)\right]\\ V_{4}=E\left[v_{4}\left(t\right)\right]=\frac{1}{4}E\left[{\hat{V}}_{1}^{2}\left(t\right)\right]+\frac{1}{4}E\left[V_{2}^{2}\left(t\right)\right]+\frac{1}{2}E\left[{\hat{V}}_{1}\left(t\right)V_{2}\left(t\right)\right]\\ V_{5}=E\left[v_{5}\left(t\right)\right]=\frac{1}{4}E\left[{\hat{V}}_{1}^{2}\left(t\right)\right]+\frac{1}{4}E\left[{\hat{V}}_{2}^{2}\left(t\right)\right]+\frac{1}{2}E\left[{\hat{V}}_{1}\left(t\right){\hat{V}}_{2}\left(t\right)\right]\\ V_{6}=E\left[v_{6}\left(t\right)\right]=\frac{1}{4}E\left[V_{1}^{2}\left(t\right)\right]+\frac{1}{4}E\left[{\hat{V}}_{2}^{2}\left(t\right)\right]-\frac{1}{2}E\left[V_{1}\left(t\right)V_{2}\left(t\right)\right] \end{array}$$

In the application of the correlation radiometer, the received signal can be regarded as a Gaussian white noise signal with a mean value of 0, and then, according to the property of Hilbert transform, the real and imaginary parts of the correlator can be obtained by differential amplification of $V_{3}\sim V_{6}$ (assuming the gain is 1)
(3)$$\begin{array}{l} V_{real}=V_{5}-V_{6}=E\left[V_{1}\left(t\right)V_{2}\left(t\right)\right]\\ V_{imag}=V_{4}-V_{3}=E\left[{\hat{V}}_{1}\left(t\right)V_{2}\left(t\right)\right] \end{array}$$

In addition, when $V_{1}\left(t\right)$ and $V_{2}\left(t\right)$ are represented by point frequency signals $a\cos\left(\omega t+\theta_{1}\right)$ and $b\cos\left(\omega t+\theta_{2}\right)$, respectively, the output of the correlator can be expressed as
(4)$$\begin{array}{c} V_{real}=\frac{1}{2}ab\cos\left(\theta_{1}-\theta_{2}\right)\\ V_{imag}=\frac{1}{2}ab\sin\left(\theta_{1}-\theta_{2}\right) \end{array}$$

From Equation (4), we can see that the output of the analog complex correlator will form a correlation circle with the phase difference of two input signals varying from 0° to 360°, and the performance of the analog correlator can be evaluated through the correlation circle.

### 2.2. Influence of Detector Mismatch on Correlation Results

In order to analyze the effect of detector mismatch on the correlator, assume that the reflection coefficient of each port in the six-port circuit is the same and expressed by $\Gamma_{s}$; the coupling coefficient of each output port is the same and expressed by $\Gamma_{c}$; the reflection coefficient of the detection circuit is expressed by $\Gamma_{D}$; the phase shift of the transmission line between the six-port network and the detector is expressed by $\theta_{l}$. If multiple coupling and reflection are ignored, the signals of each channel entering the detector can be expressed as
(5)$$\begin{array}{c} {S^{\prime }}_{3}=\frac{1}{2}\left[\left(1+\Gamma_{D}\Gamma_{S}e^{j2\theta_{L}}\right)\sqrt{1-\Gamma_{D}^{2}}S_{3}+\sqrt{\left(1-\Gamma_{D}^{2}\right)\left(1-\Gamma_{S}^{2}\right)}\Gamma_{D}\Gamma_{C}\left(S_{4}+S_{5}+S_{6}\right)\right]\\ {S^{\prime }}_{4}=\frac{1}{2}\left[\left(1+\Gamma_{D}\Gamma_{S}e^{j2\theta_{L}}\right)\sqrt{1-\Gamma_{D}^{2}}S_{4}+\sqrt{\left(1-\Gamma_{D}^{2}\right)\left(1-\Gamma_{S}^{2}\right)}\Gamma_{D}\Gamma_{C}\left(S_{3}+S_{5}+S_{6}\right)\right]\\ {S^{\prime }}_{5}=\frac{1}{2}\left[\left(1+\Gamma_{D}\Gamma_{S}e^{j2\theta_{L}}\right)\sqrt{1-\Gamma_{D}^{2}}S_{5}+\sqrt{\left(1-\Gamma_{D}^{2}\right)\left(1-\Gamma_{S}^{2}\right)}\Gamma_{D}\Gamma_{C}\left(S_{3}+S_{4}+S_{6}\right)\right]\\ {S^{\prime }}_{6}=\frac{1}{2}\left[\left(1+\Gamma_{D}\Gamma_{S}e^{j2\theta_{L}}\right)\sqrt{1-\Gamma_{D}^{2}}S_{6}+\sqrt{\left(1-\Gamma_{D}^{2}\right)\left(1-\Gamma_{S}^{2}\right)}\Gamma_{D}\Gamma_{C}\left(S_{3}+S_{4}+S_{5}\right)\right] \end{array}$$

The four input signals pass through a square-law detector with a second-order voltage conversion coefficient of 1 and pass through LPF to filter out the high-frequency term, and then the DC output can be expressed as
(6)$$\begin{array}{cc} {V^{\prime }}_{3} & =\frac{1}{4}{\left(1+\Gamma_{D}\Gamma_{S}e^{j2\theta_{L}}\right)}^{2}\left(1-\Gamma_{D}^{2}\right)E\left(S_{3}^{2}\right)+\frac{1}{4}\left(1-\Gamma_{D}^{2}\right)\left(1-\Gamma_{S}^{2}\right)\Gamma_{D}^{2}\Gamma_{C}^{2}E\left[{\left(S_{4}+S_{5}+S_{6}\right)}^{2}\right]\\ & +\frac{1}{2}\left(1+\Gamma_{D}\Gamma_{S}e^{j2\theta_{L}}\right)\sqrt{1-\Gamma_{S}^{2}}\Gamma_{D}\Gamma_{C}\left(1-\Gamma_{D}^{2}\right)E\left[S_{3}\left(S_{4}+S_{5}+S_{6}\right)\right]\\ {V^{\prime }}_{4} & =\frac{1}{4}{\left(1+\Gamma_{D}\Gamma_{S}e^{j2\theta_{L}}\right)}^{2}\left(1-\Gamma_{D}^{2}\right)E\left(S_{4}^{2}\right)+\frac{1}{4}\left(1-\Gamma_{D}^{2}\right)\left(1-\Gamma_{S}^{2}\right)\Gamma_{D}^{2}\Gamma_{C}^{2}E\left[{\left(S_{3}+S_{5}+S_{6}\right)}^{2}\right]\\ & +\frac{1}{2}\left(1+\Gamma_{D}\Gamma_{S}e^{j2\theta_{L}}\right)\sqrt{1-\Gamma_{S}^{2}}\Gamma_{D}\Gamma_{C}\left(1-\Gamma_{D}^{2}\right)E\left[S_{4}\left(S_{3}+S_{5}+S_{6}\right)\right]\\ {V^{\prime }}_{5} & =\frac{1}{4}{\left(1+\Gamma_{D}\Gamma_{S}e^{j2\theta_{L}}\right)}^{2}\left(1-\Gamma_{D}^{2}\right)E\left(S_{5}^{2}\right)+\frac{1}{4}\left(1-\Gamma_{D}^{2}\right)\left(1-\Gamma_{S}^{2}\right)\Gamma_{D}^{2}\Gamma_{C}^{2}E\left[{\left(S_{3}+S_{4}+S_{6}\right)}^{2}\right]\\ & +\frac{1}{2}\left(1+\Gamma_{D}\Gamma_{S}e^{j2\theta_{L}}\right)\sqrt{1-\Gamma_{S}^{2}}\Gamma_{D}\Gamma_{C}\left(1-\Gamma_{D}^{2}\right)E\left[S_{5}\left(S_{3}+S_{4}+S_{6}\right)\right]\\ {V^{\prime }}_{6} & =\frac{1}{4}{\left(1+\Gamma_{D}\Gamma_{S}e^{j2\theta_{L}}\right)}^{2}\left(1-\Gamma_{D}^{2}\right)E\left(S_{6}^{2}\right)+\frac{1}{4}\left(1-\Gamma_{D}^{2}\right)\left(1-\Gamma_{S}^{2}\right)\Gamma_{D}^{2}\Gamma_{C}^{2}E\left[{\left(S_{3}+S_{4}+S_{5}\right)}^{2}\right]\\ & +\frac{1}{2}\left(1+\Gamma_{D}\Gamma_{S}e^{j2\theta_{L}}\right)\sqrt{1-\Gamma_{S}^{2}}\Gamma_{D}\Gamma_{C}\left(1-\Gamma_{D}^{2}\right)E\left[S_{6}\left(S_{3}+S_{4}+S_{5}\right)\right] \end{array}$$

Therefore, the output of the real part and the imaginary part of the complex correlator can be expressed as
(7)$$\begin{array}{cc} {V^{\prime }}_{real} & =\left[{\left(1+\Gamma_{D}\Gamma_{S}e^{j2\theta_{L}}\right)}^{2}-2\left(1-\Gamma_{S}^{2}\right)\Gamma_{D}^{2}\Gamma_{C}^{2}+\left(1+\Gamma_{D}\Gamma_{S}e^{j2\theta_{L}}\right)\sqrt{1-\Gamma_{S}^{2}}\Gamma_{D}\Gamma_{C}\right]\left(1-\Gamma_{D}^{2}\right)E\left[\left(V_{1}\left(t\right)V_{2}\left(t\right)\right)\right]\\ & +\left[\left(1-\Gamma_{S}^{2}\right)\Gamma_{D}^{2}\Gamma_{C}^{2}-\left(1+\Gamma_{D}\Gamma_{S}e^{j2\theta_{L}}\right)\sqrt{1-\Gamma_{S}^{2}}\Gamma_{D}\Gamma_{C}\right]\left(1-\Gamma_{D}^{2}\right)\left\{E\left[\left({\hat{V}}_{1}\left(t\right)V_{2}\left(t\right)\right)\right]-E\left[V_{2}^{2}\left(t\right)\right]\right\}\\ {V^{\prime }}_{imag} & =\left[{\left(1+\Gamma_{D}\Gamma_{S}e^{j2\theta_{L}}\right)}^{2}-2\left(1-\Gamma_{S}^{2}\right)\Gamma_{D}^{2}\Gamma_{C}^{2}+\left(1+\Gamma_{D}\Gamma_{S}e^{j2\theta_{L}}\right)\sqrt{1-\Gamma_{S}^{2}}\Gamma_{D}\Gamma_{C}\right]\left(1-\Gamma_{D}^{2}\right)E\left[\left({\hat{V}}_{1}\left(t\right)V_{2}\left(t\right)\right)\right]\\ & +\left[\left(1-\Gamma_{S}^{2}\right)\Gamma_{D}^{2}\Gamma_{C}^{2}-\left(1+\Gamma_{D}\Gamma_{S}e^{j2\theta_{L}}\right)\sqrt{1-\Gamma_{S}^{2}}\Gamma_{D}\Gamma_{C}\right]\left(1-\Gamma_{D}^{2}\right)\left\{E\left[\left(V_{1}\left(t\right)V_{2}\left(t\right)\right)\right]-E\left[V_{1}^{2}\left(t\right)\right]\right\} \end{array}$$

According to the above formula, the following discussion can be carried out. Firstly, assuming that the six-port network is ideal, which means $\Gamma_{s}=\Gamma_{c}=0$, then Equation (7) can be simplified as
(8)$$\begin{array}{c} {V^{\prime }}_{real}=\left(1-\Gamma_{D}^{2}\right)E\left[\left(V_{1}\left(t\right)V_{2}\left(t\right)\right)\right]\\ {V^{\prime }}_{imag}=\left(1-\Gamma_{D}^{2}\right)E\left[\left({\hat{V}}_{1}\left(t\right)V_{2}\left(t\right)\right)\right] \end{array}$$

From Equation (8), we can see that the output of the correlator is the same as that of Equation (3). The only difference is the decline in correlator output caused by the reflection of the sensor. In this case, the reflected signal of the detection circuit will be absorbed by the 50-ohm load resistance or the internal resistance of the signal source after passing through the six-port network, and the reflected signal does not affect the correlator. In the case that the six-port network is not ideal, the input and output ports of the six-port network have reflection and coupling between them. The reflected signals from the detector input will form crosstalk inside the six-port network, thus affecting the output of the correlator. Therefore, the analysis of detector mismatch should be established in the case that the six-port network is not ideal.

Secondly, assuming that the two input signals are uncorrelated, then Equation (7) can be simplified as
(9)$$\begin{array}{c} {V^{\prime }}_{real}=\left[\left(1+\Gamma_{D}\Gamma_{S}e^{j2\theta_{L}}\right)\sqrt{1-\Gamma_{S}^{2}}\Gamma_{C}-\left(1-\Gamma_{S}^{2}\right)\Gamma_{C}^{2}\right]\left(1-\Gamma_{D}^{2}\right)•E\left[V_{2}^{2}\left(t\right)\right]\\ {V^{\prime }}_{imag}=\left[\left(1+\Gamma_{D}\Gamma_{S}e^{j2\theta_{L}}\right)\sqrt{1-\Gamma_{S}^{2}}\Gamma_{C}-\left(1-\Gamma_{S}^{2}\right)\Gamma_{C}^{2}\right]\left(1-\Gamma_{D}^{2}\right)•E\left[V_{1}^{2}\left(t\right)\right] \end{array}$$

It shows that, even if the input signal is uncorrelated, the output of the correlator is still not zero due to the reflection of the detection circuit and the reflection and coupling of the six-port network. In addition, the phase shift $\theta_{l}$ of the transmission line between the detector and the six-port network will also affect the output of the correlator. The specific impact will be explained by the following simulation by establishing simulation model in ADS (Advanced Design System), which is shown in Figure 2.

Firstly, when the isolation and return loss of each port in the six-port network are fixed at 20 dB and the reverse isolation is fixed at 20 dB, the influence of the phase shift of the transmission line of the detector and the six-port network on the correlation results can be simulated.

Figure 3 shows the simulation results of the variation in the correlation circle and its radius with the phase shift $\theta_{l}$. It is worth noting that the ideal case in the figure refers to the ideal reverse isolation between the detector and the six-port network, and we can see that, with the change in $\theta_{l}$, the radius of the correlation circle will increase or decrease. This is due to the crosstalk caused by the reflected signal in the correlator, not the actual correlation result between the two input signals. Therefore, the increase or decrease in the correlation circle radius should be avoided, and it is necessary to increase the reverse isolation between the six-port network and the detector. The demand for reverse isolation degree can be determined when $\theta_{l}$ has the greatest influence, that is, when $\theta_{l}$ is 15° and 105°, respectively. Therefore, the simulation results of the variation in correlation circle radius with the reverse isolation under the condition that the $\theta_{l}$ is 15° and 105°, respectively, are shown in Figure 4.

It can be seen that, under different $\Gamma_{c}\&\Gamma_{s}$, reverse isolation has different influence on the radius of the correlation circle. Since the return loss of the six-port integrated chip is about 10 dB [33], the corresponding $\Gamma_{c}\&\Gamma_{s}$ is 0.316. If we take a 2% deterioration in the radius of the correlation circle as the evaluation criterion, then the reverse isolation needs to be greater than 30 dB.

### 2.3. Analysis of Transmission Line between Six-Port and Detector on Correlation Results

In the simulation results in Figure 3, we can also find an interesting conclusion: when the reverse isolation between the detector and the six-port network is ideal, the phase shift $\theta_{l}$ of the transmission line between them has no effect on the relevant results.

Assuming that the length of transmission lines between the six-port network and the detector is different, these transmission lines are lossless, and the schematic diagram is shown in Figure 5. Further, when two input signals, $S_{1}$ and $S_{2}$, are represented by $\cos\left(\omega t+\theta_{1}\right)$ and $\cos\left(\omega t+\theta_{2}\right),$ respectively, then the four outputs of the six-port network can be expressed as
(10)$$\begin{array}{l} S_{3}=\frac{1}{2}\left[\cos\left(\omega t+\theta_{1}+\theta_{l3}\right)+\sin\left(\omega t+\theta_{2}+\theta_{l3}\right)\right]\\ S_{4}=\frac{1}{2}\left[\sin\left(\omega t+\theta_{1}+\theta_{l4}\right)+\cos\left(\omega t+\theta_{2}+\theta_{l4}\right)\right]\\ S_{5}=\frac{1}{2}\left[\sin\left(\omega t+\theta_{1}+\theta_{l5}\right)+\sin\left(\omega t+\theta_{2}+\theta_{l5}\right)\right]\\ S_{6}=\frac{1}{2}\left[-\cos\left(\omega t+\theta_{1}+\theta_{l6}\right)+\cos\left(\omega t+\theta_{2}+\theta_{l6}\right)\right] \end{array}$$
where $\theta_{li}$ (i=3~6) represents the phase shift of the four-way transmission line. Take the signal of port 3 as an example; its output after passing through the square-law detector can be expressed as
(11)$$\begin{array}{cc} v_{3} & =\frac{1}{4}{\left[\cos\left(\omega t+\theta_{1}+\theta_{l3}\right)+\sin\left(\omega t+\theta_{2}+\theta_{l3}\right)\right]}^{2}\\ & =\frac{1}{8}\left[1+\cos\left(2\omega t+2\theta_{1}+2\theta_{l3}\right)\right]+\frac{1}{8}\left[1-\cos\left(2\omega t+2\theta_{2}+2\theta_{l3}\right)\right]\\ & +\frac{1}{4}\left[\sin\left(2\omega t+\theta_{1}+\theta_{2}+2\theta_{l3}\right)-\sin\left(\theta_{1}-\theta_{2}\right)\right] \end{array}$$

After the signal passes through the LPF and filters out the high-frequency term, the DC output can be expressed as
(12)$$V_{3}=\frac{1}{8}-\frac{1}{4}\sin\left(\theta_{1}-\theta_{2}\right)$$

It can be seen that the DC signal output after detection and integration is independent of the length of the transmission line. Therefore, when the isolation between the six-port and the detector is ideal and the transmission line loss is not considered, the length of the transmission line between the six-port and the detector does not affect the correlation results. Therefore, in the design of the correlator, under the premise of ensuring the isolation between the six-port network and the detector, the length of the transmission line between them can be different.

## 3. Design of the Ka band Analog Complex Correlator

According to the analysis in Section 2, the schematic diagram of the designed analog complex correlator with the operating frequency of 32 to 36 GHz is illustrated in Figure 6. Further, we can see that the designed correlator is made up of three parts: the integrated six-port chip, four amplification circuits, and four detectors. Furthermore, the integrated six-port chip integrates the functions of amplification, phase shifting, and six-port signal distribution network. Behind the integrated chip are four amplifier chips; using the reverse isolation characteristics of transistors, amplifiers can provide good reverse isolation between the six-port network and detectors and can also amplify the input radio frequency (RF) signals. Finally, four square-law detectors perform a square operation on the input signal and filter the high-frequency signal through the LPF, resulting in a DC signal containing the correlation results.

Next, we will briefly introduce the basic situation of the six-port chip, then design the square-law detector, and finally complete the integration and implementation of the designed analog correlator.

### 3.1. Introduction of the Integrated Six-Port Chip

The six-port chip that is used in the correlator has been completed in our previous work [33]. The six-port chip is designed and fabricated using the 0.15 um GaAs process, and the size of the final fabricated chip, which is shown in Figure 7, is 5 mm × 2 mm.

From Figure 7, we can see that the six-port chip integrates two amplifiers with three-stage transistors, two phase shifters with low-pass and high-pass structures, and a six-port network. The input signals will enter ports 1 and 2 of the chip, then pass through amplifiers and phase shifters; finally, the signals are distributed by the six-port network and output from ports 3 to 6. For the designed six-port chip, two amplifiers need 3 V DC voltage with the total current of 73 mA. In addition, the phase shift range of a single phase shifter is greater than 180° and less than 360°. Therefore, two phase shifters in the six-port chip need to work together to ensure that the phase difference of two input signals can be changed in the range of 0° to 360°.

### 3.2. Design and Measurement of the Detector

The design of the detector includes the selection of the detector diode, the design of output LPF, and the design of the input matching network [37]. In this paper, the zero-bias Schottky diode produced by Skyworks is selected as the detection diode, and the model is SMSA7630-061. The use of a zero-bias diode can avoid DC biasing of the detector output and reduce the complexity of the detection circuit. The LPF uses microstrip branches to suppress the input RF signal, and the simulation and measurement results of the LPF are shown in Figure 8, and the suppression of LPF to the fundamental signal (32 to 36 GHz) is greater than 38 dB. It should be noted that this filter is only used for filtering RF signals, so an RC low-pass filter needs to be added later to adjust the integration time of the correlator.

Since the parameters of the detector diode are not accurate in the Ka band, it is necessary to obtain the matching state of the detector diode before designing the input matching network of the detector. Then, the calibration model according to TRL (through, reflect, and line) calibration has been designed and measured [38], and the test structure is also designed to measure the matching state of the detector diode. The TRL calibration model and test structure are shown in Figure 9a, and the measurement and calibration results of the test structure are shown in Figure 9b. After the calibration, the design of the input matching network can be completed based on the matching points of the diodes on the Smith chart. In order to obtain higher voltage sensitivity, the input matching network adopts the matching mode of lossless microstrip branch. Then, the final fabricated detector and its measurement results are shown in Figure 9c, and we can see that the return loss of the detector is better than 7.5 dB when the working frequency is 32 to 36 GHz. Figure 9d shows the output voltage of the detector versus input power, which indicates that the input power of the detector should be less than −6 dBm to ensure the detector operates within the square-law detection range. Moreover, with the input power of −25 dBm, the voltage sensitivity of the detector can be calculated to be 0.8 V/mw. It should be noted that, if the matching state of the detector does not meet the requirements for the reverse isolation between six-port and detector, then the amplifier chip is necessary to avoid the impact of the reflected signal from the detector on the correlator.

### 3.3. Integration of the Analog Correlator

The integration of the Ka band analog complex correlator can be completed based on the integrated six-port chip and the detection circuit, which have been designed and measured, and the final fabricated analog complex correlator is illustrated in Figure 10. In order to prevent the DC signal generated by the detector from entering the six-port network, the band pass filter (BPF), which plays the role of suppressing clutter and DC block, is added between the detector and amplifier chip.

## 4. Measurement of the Ka band Analog Complex Correlator

The measurement of the analog correlator includes the test of the dynamic range of input power, correlation efficiency, and noise characteristics. The dynamic range of input power and correlation efficiency can be measured by using a vector network analyzer to provide two channel single-frequency signals with equal amplitude and controllable phase. Then, the block diagram and the test site of the single-frequency measurement are shown in Figure 11.

It can be seen that both the integrated six-port chip and the amplifier chip in the correlator require the 3 V DC power supply, which is provided by the Rohde and Schwarz HMP 2030 programmable power supply, and the total current is 232 mA. Furthermore, the phase shifting function in the six-port chip requires a variable DC voltage of 0 to 1.5 V to complete the phase control of the input signals, which is also provided by the programmable power supply. The variation in the output voltage provided by the DC power supply can be controlled by the computer, and the DC voltage output by the correlator is also collected by a digital multimeter and then transmitted to the computer. This method completes the semi-automatic measurement of the correlator and obtains a series of correlation circles.

In actual measurement results, the measured correlation circle is not ideal and will be accompanied by a series of errors [35], so calibration is needed to eliminate the measurement errors, and the calibration model [36] is shown as:(13)$$\begin{array}{l} V_{real}=ab\cos\left(\theta_{1}-\theta_{2}\right)+C_{r}\\ V_{imag}=abg\sin\left(\theta_{1}-\theta_{2}+\phi \right)+C_{i} \end{array}$$
where g is the quadrature amplitude error of the correlator and ϕ is the quadrature phase error, $C_{r}$ and $C_{i}$ are the DC offset error of the correlator. Then, these error parameters can be calculated through the least squares method.

### 4.1. Measurement of Dynamic Range of Input Power

In order to measure the dynamic range of the input power for the designed correlator, the frequency of the input signal should be fixed at the center frequency (34 GHz), and the phase difference of the input signals is controlled by the vector network analyzer. Correspondingly, the control voltage of two phase shifters is set to 0 V and remains unchanged. When the power of the input signal changes from −30 dBm to −50 dBm, the measured correlation circle is represented in Figure 12a, and the calibration results of the correlation circle are represented in Figure 12b–d.

Figure 12c shows the variation in the correlation circle radius with input power. By fitting the measurement curve with a straight line, we can see that, when the input power of the correlator is less than −35 dBm, the output voltage of the correlator is proportional to the input power, and the detector operates in the square-law detection region. When the input power of the correlator is greater than −35 dBm, the input power of the detector exceeds the square-law detection range, which is inappropriate for the correlator, so the input signal power of the designed correlator should be less than −35 dBm. Further, from Figure 12d, the maximum phase detection error is less than 2.8° when the power of the input signal is less than −35 dBm. In addition, using the calibrated data, we can also know that the quadrature amplitude error of the designed correlator is less than ±0.7 dB, and the quadrature phase error is less than ±2°.

In the above measurement, the maximum input power of the correlator is obtained. Next, it is necessary to measure the minimum input power of the correlator. According to the measurement scheme in Figure 11, and by amplifying the DC output voltage of the correlator with a differential amplifier, the test results of the correlator at lower input power can be obtained. When the input power varies from −70 to −50 dBm, the correlation circle can be obtained by 500 times differential amplification of the four output voltage signals of the correlator. The measurement and calibration results are represented in Figure 13.

It can be seen that the measured correlation circle fluctuates and has a –1 V DC bias, which is caused by the differential amplifier. The DC bias can be eliminated by calibration, and the fluctuation of the correlation circle will cause the maximum phase detection error to deteriorate to 5°. Moreover, it can be found in Figure 13c that the output voltage of the correlator is proportional to the input power, which means that, when the correlator is used in the field of temperature measurement, the power of the input signal can be as low as –70 dBm. To sum up, the dynamic range of input power of the Ka band analog complex correlator is −70 dBm to −35 dBm.

### 4.2. Measurement of the Correlation Efficiency and Equivalent Bandwidth

The correlation efficiency and the equivalent bandwidth of the analog correlator can be calculated using the amplitude and phase characteristics of the correlator’s passband. Essentially, the correlation efficiency reflects the working bandwidth utilization of the correlator. Therefore, the expression of the correlation efficiency and the equivalent bandwidth of the correlator are similar, and they are formulated as [16,19]:(14)$$\eta =\frac{{\left|\int \left|W\left(f\right)\right|\cos\left[\delta \left(f\right)\right]df\right|}^{2}}{\int B{\left|W\left(f\right)\right|}^{2}df}$$
(15)$$B_{e}=\frac{{\left|\int W\left(f\right)df\right|}^{2}}{\int {\left|W\left(f\right)\right|}^{2}df}$$
where η means the correlation efficiency of the analog correlator, |W(f)| is the amplitude response of the correlator, and cos[δ(f)] is the phase response. Moreover, B and $B_{e}$ are the working bandwidth and equivalent bandwidth. It should be noted that the correlation efficiency of the analog correlator needs to be greater than 90% [19,37].

In order to measure the correlation efficiency and the equivalent bandwidth of the designed correlator, the power of the input signal is −35 dBm, and the phase difference of the two input signals is controlled by the phase shifter in the six-port chip. Then, the measurement and calibration results are represented in Figure 14, with the frequency of the input signal varying from 30 GHz to 38 GHz.

The phase and amplitude response of the correlator within the operating bandwidth is needed to calculate the correlation efficiency and equivalent bandwidth. The amplitude response of the correlator can be equivalent to the normalized fluctuation of the correlation circle radius, and the phase response can be equivalent to the maximum phase detection error of each frequency point. Through this method, the final calculated correlation efficiency is 92.5% and the equivalent bandwidth is 3.42 GHz. Then, some important characteristics of the designed correlator are compared with others’ work, which is represented in Table 1. It is obvious that the proposed analog correlator has the highest working frequency, and the correlation efficiency is deteriorated, which is caused by the fluctuation in the output of the correlator in the passband and the deterioration in the maximum phase detection error. Furthermore, the deterioration in the maximum phase detection error is caused by the phase shift accuracy of phase shifter in six-port chip.

### 4.3. Broadband Noise Measurement

In practical application, the correlator needs to complete the correlation calculation on the broadband noise signal. Therefore, the designed correlator needs to be measured when the input signal is a noise signal to observe the characteristics of the analog correlator in the actual working environment. Therefore, the measurement platform for correlator noise characteristics evaluation has been designed and shown in Figure 15.

It can be seen that the correlated noise signal is generated by the noise source, and the output of the noise source is followed by a variable attenuator, which is used to change the power of the input noise signal. Then, a power divider is adopted to split the noise signal into two outputs, which can be used as two input noise signals of the analog correlator. In addition, the phase difference of two input noise signals can be controlled through the integrated six-port chip. Finally, the DC output voltage of the correlator is collected by a digital multimeter and the data are transmitted to the computer for processing.

During the measurement, the phase difference of two input signals varies from −180° to 180°, and the power of the noise signal varies from −51 dBm to −31 dBm. Then, the measurement and calibration results for different input power are shown in Figure 16.

It can be seen that, in the case of broadband noise signal input, the output voltage of the correlator is proportional to the input noise power. In addition, the maximum phase detection error is also deteriorated, which is not only due to the impact of the phase shift control accuracy of the phase shifter but also affected by the phase shift of the broadband noise signal.

## 5. The Millimeter-Wave Radiation Temperature Measurement System Based on the Designed Analog Correlator

The millimeter-wave radiation temperature measurement system is based on the designed analog complex correlator, and the block diagram is shown in Figure 17.

The external noise signal is received by the horn antenna and converted to the microstrip line through the waveguide microstrip conversion structure. Then, the noise signal passes through the 180° hybrid bridge and is added or subtracted from the noise signal generated by the reference resistance to obtain two signals combined by external noise and internal resistance noise. If the equivalent noise temperature of the noise signal received by the antenna and the noise signal generated by the reference resistance is expressed as $T_{A}$ and $T_{ref}$ respectively, the output of the two noise signals after passing through the 180 hybrid bridge can be expressed as
(16)V1=22(VTA+VTref)V2=22(VTA−VTref)

Then, two signals enter the analog complex correlator, and output of the correlator can be expressed as
(17)$$\begin{array}{l} V_{real}=G\left(T_{A}-T_{ref}\right)\cos\theta \\ V_{imag}=G\left(T_{A}-T_{ref}\right)\sin\theta \end{array}$$
where *G* represents the gain coefficient, we can see that the output of the correlator is proportional to the difference between antenna temperature $T_{A}$ and reference temperature $T_{ref}$. Then, the DC signal output by the correlator is transmitted to the system terminal for processing after passing through the amplification and digital acquisition circuit. According to the block diagram in Figure 17, the final designed thermometer is shown in Figure 18.

It can be seen that the size of the fabricated thermometer is 140 mm × 47 mm × 20 mm, which meets the handheld demand of human body temperature measurement. In addition, Figure 18 only shows the front side of the thermometer and only completes the data amplification and sampling of correlation results. The subsequent data processing and display functions are integrated on the digital board into the back of the thermometer.

Since the input power required by the analog correlator can be as low as −70 dBm, only one low-noise amplifier (LNA) in the thermometer can meet the input power demand of the analog correlator, which effectively reduces the influence of gain fluctuation on the thermometer output voltage. Furthermore, it can be seen that, shown in Figure 18, the length of the transmission line between the six-port network and the detector is different, which has no effect on the output of the correlator according to the previous analysis.

For the measurement of the thermometer, the noise source is used as the input of the thermometer. By changing the input power of the noise signal, the equivalent noise temperature can be calculated, and then the output voltage of the designed thermometer at different input temperatures is measured. The test site of the thermometer is represented in Figure 19, and the measurement results of the thermometer are represented in Figure 20.

Figure 20a shows the measurement results of the thermometer output voltage at 12 different equivalent noise temperatures, and ten measurements are completed at each equivalent noise temperature to determine the fluctuation of the thermometer output voltage at a fixed input temperature. Then, the variation in the average output voltage of the thermometer with the equivalent input noise temperature is shown in Figure 20b, and the standard deviation of the thermometer output voltage is shown in Figure 20c. It can be seen that the standard deviation of the thermometer is less than 0.6 mV at all input temperatures.

The temperature sensitivity of the thermometer can be calculated according to the slope of the output voltage changing with the input temperature in Figure 20b and the standard deviation in Figure 20c. Then, the thermometer temperature sensitivity is shown in Figure 20d, and we can see that, when the input noise temperature is less than 150 °C, the temperature sensitivity of the thermometer is less than 0.2 K. For the application of human body temperature measurement, the input temperature is within the range of 25 °C to 40 °C, so the temperature sensitivity of the thermometer is less than 0.2 K.

## 6. Conclusions

In this paper, a Ka band (32 to 36 GHz) add-and-square analog correlator using an integrated six-port chip is presented for the millimeter-wave radiation temperature measurement system. The designed correlator adopts the six-port technique to obtain broadband characteristics. Then, a six-port chip, which integrates the functions of amplification, phase shifting, and six-port signal distribution network, is used to miniaturize the correlator, and the size of the six-port chip is 5 mm × 2 mm. In addition, the analysis and simulation prove that it is necessary to increase the reverse isolation between the six-port network and the detector. Based on the above conclusions, an analog correlator has been designed, fabricated, and measured. From the measurement results, we can see that the dynamic range of the input power is −70 dBm to −35 dBm. Moreover, the correlation efficiency of the designed correlator is 92.5%, and the equivalent bandwidth is 3.42 GHz. According to the broadband noise measurement, the output voltage of the correlator is proportional to the input noise power, which reveals that the correlator is suitable for the field of human body temperature measurement. Finally, a millimeter-wave radiation temperature measurement system based on the Ka band correlator is also designed. The measurement results reveal that the temperature sensitivity of the thermometer for human body temperature measurement is less than 0.2 K.

In the next step, it is necessary is to complete the calibration of the thermometer and test the actual body temperature. In addition, it is also necessary to solve the difficulty of phase shift control accuracy of the phase shifter in a six-port chip to improve the phase detection error of the correlator. On the premise of completing the above work, the correlator can be used in the existing passive millimeter-wave security imaging system [39,40] to eliminate local oscillator (LO) links in the original system and reduce the volume of the system. Furthermore, passive-millimeter-wave imaging technology and optical imaging technology [41,42] can be integrated to obtain millimeter-wave and optical information of the measured human body, which is conducive to improving the detection rate of dangerous things in the security imaging system.

## Figures and Tables

**Figure 1 micromachines-14-00867-f001:**
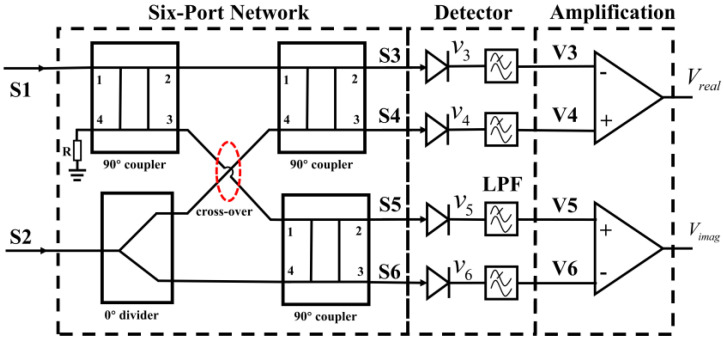
The typical block diagram of add-and-square analog correlator.

**Figure 2 micromachines-14-00867-f002:**
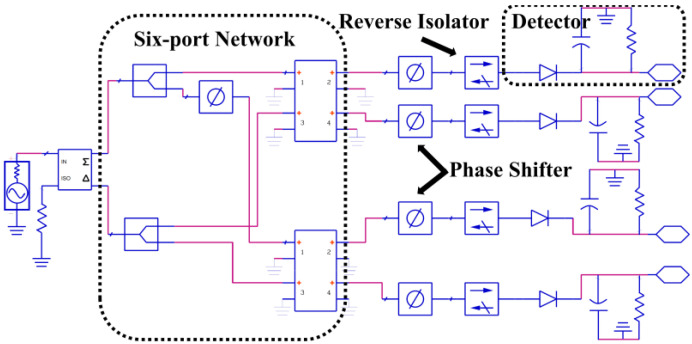
Simulation model of analog correlator in ADS.

**Figure 3 micromachines-14-00867-f003:**
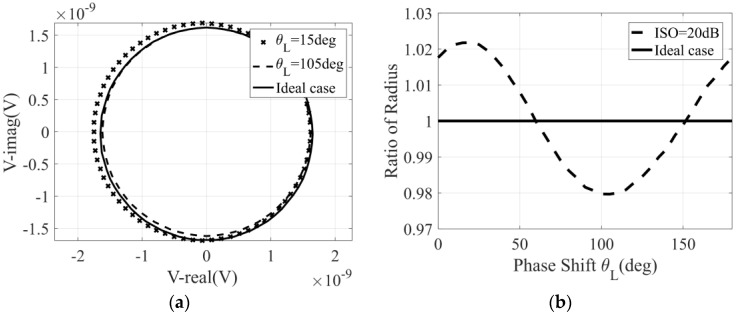
The variation in the correlation circle and its radius with the phase shift $\theta_{l}$: (**a**) correlation circle; (**b**) correlation circle radius.

**Figure 4 micromachines-14-00867-f004:**
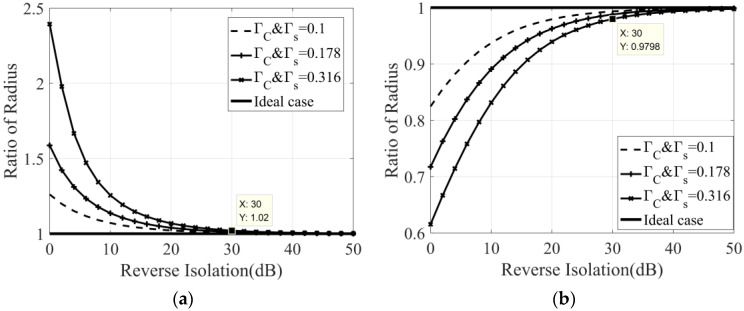
The variation in correlation circle radius with the reverse isolation: (**a**) $\theta_{l}$ is 15°; (**b**) $\theta_{l}$ is 105°.

**Figure 5 micromachines-14-00867-f005:**
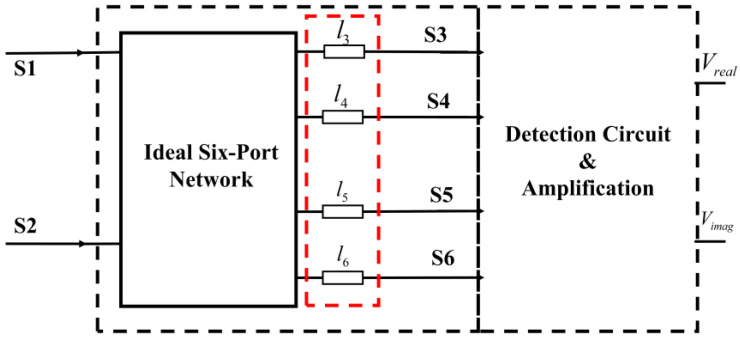
The schematic diagram of transmission line between six-port network and detector.

**Figure 6 micromachines-14-00867-f006:**
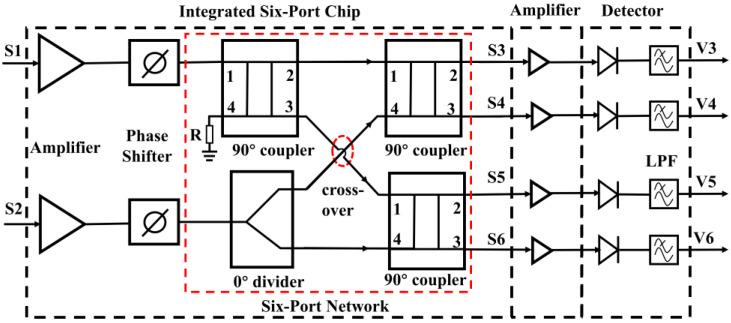
The circuit diagram of the designed correlator based on the integrated six-port chip.

**Figure 7 micromachines-14-00867-f007:**
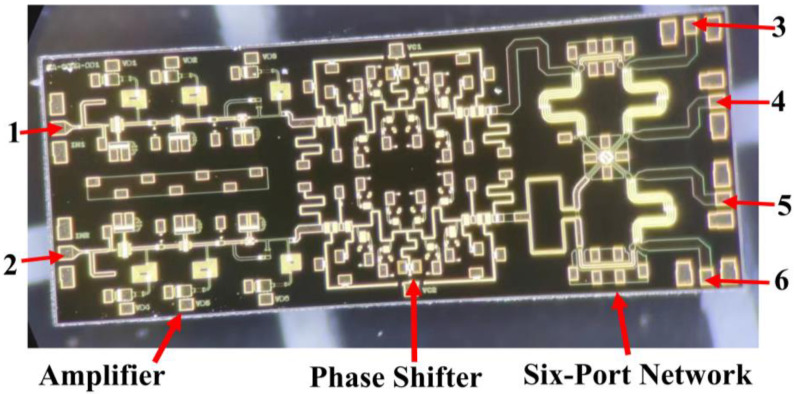
The designed six-port chip used in the analog correlator.

**Figure 8 micromachines-14-00867-f008:**
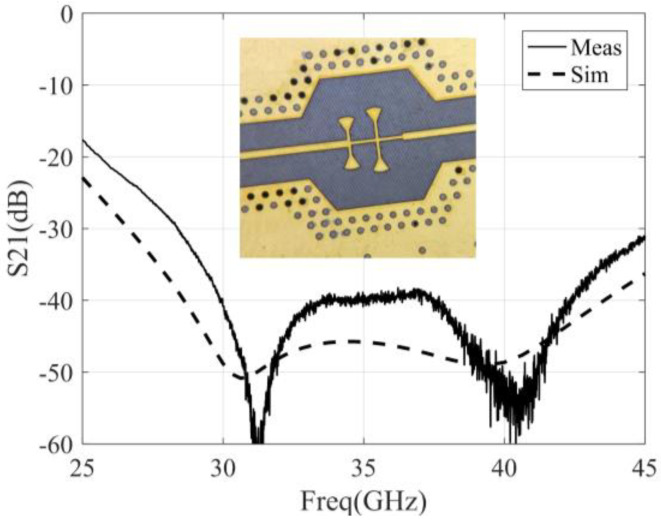
The simulation and measurement results of the LPF at the output of the detector.

**Figure 9 micromachines-14-00867-f009:**
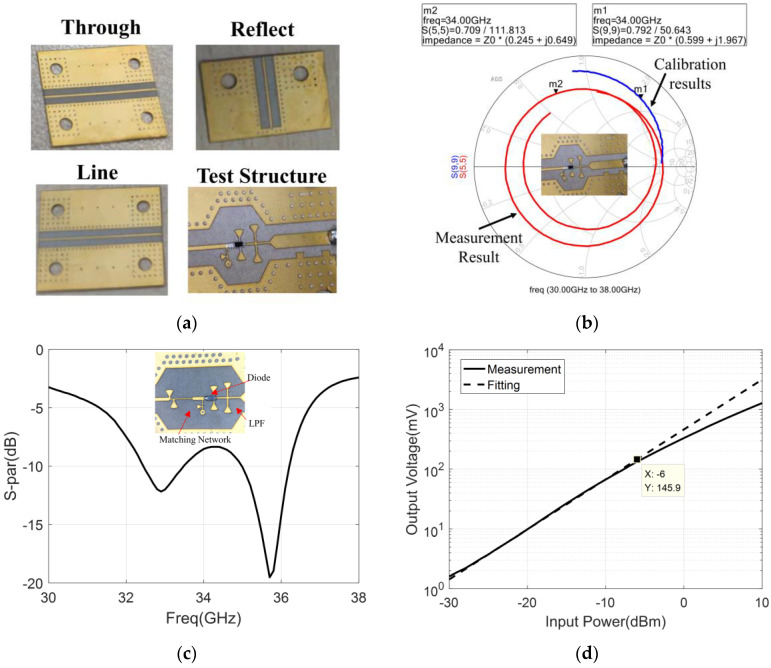
The calibration, design, and measurement of the detector: (**a**) the TRL calibration model and test structure; (**b**) the measurement and calibration results of the test structure; (**c**) the final fabricated detector and its return loss; (**d**) the output voltage of the detector versus input power.

**Figure 10 micromachines-14-00867-f010:**
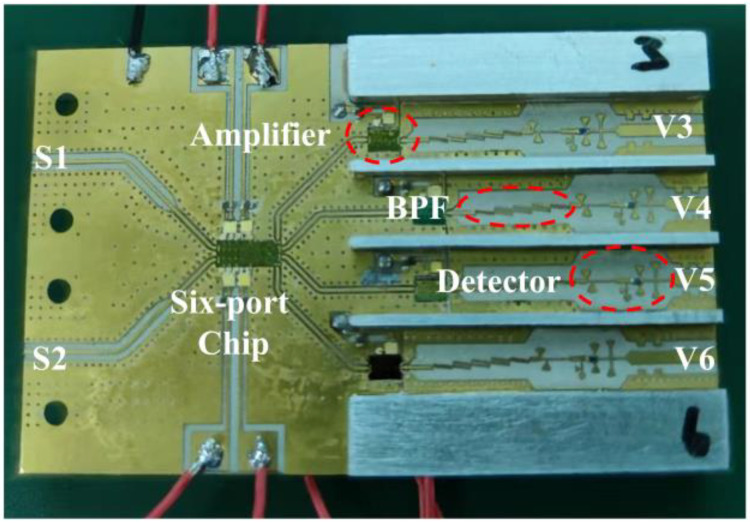
The fabricated analog correlator.

**Figure 11 micromachines-14-00867-f011:**
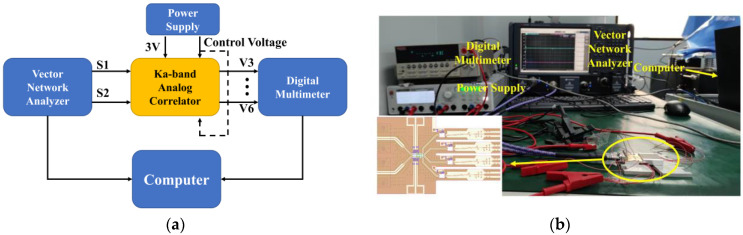
The block diagram and the test site of the single-frequency measurement: (**a**) measurement block diagram; (**b**) test site.

**Figure 12 micromachines-14-00867-f012:**
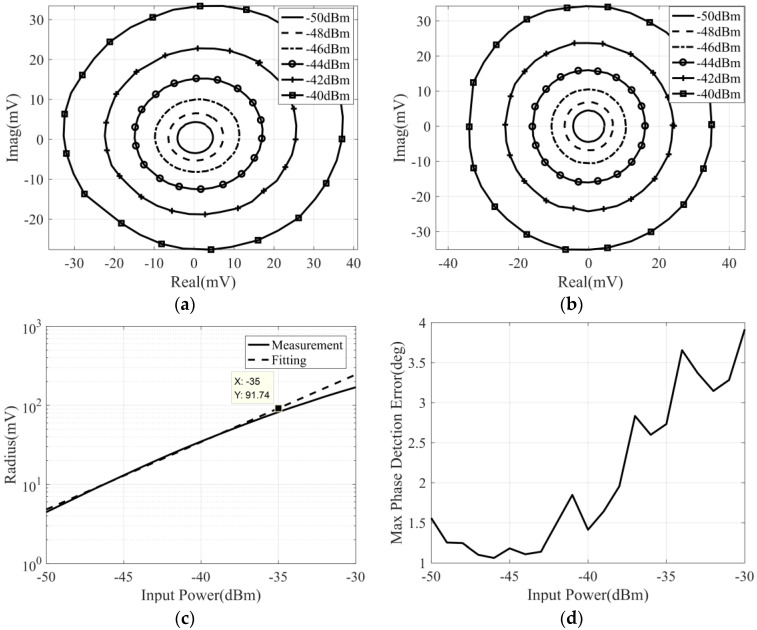
The measurement and calibration result of the correlator under the condition that the frequency of the input signals is 34 GHz and the power varies from −30 dBm to −50 dBm: (**a**) measured correlation circle; (**b**) calibrated correlation circle; (**c**) the correlation circle radius versus input power; (**d**) the variation in maximum phase detection error with input power.

**Figure 13 micromachines-14-00867-f013:**
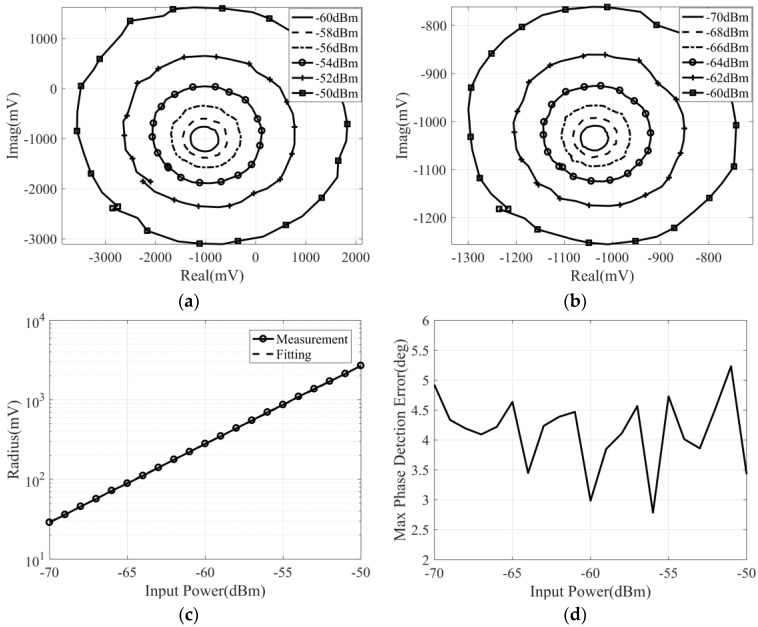
The measurement result of the correlator under the condition that the frequency of the input signals is 34 GHz and the power varies from −50 dBm to −70 dBm: (**a**) measured correlation circle with input power varies from −50 dBm to −60 dBm; (**b**) measured correlation circle with input power varies from −60 dBm to −70 dBm; (**c**) the correlation circle radius versus input power; (**d**) the variation in maximum phase detection error with input power.

**Figure 14 micromachines-14-00867-f014:**
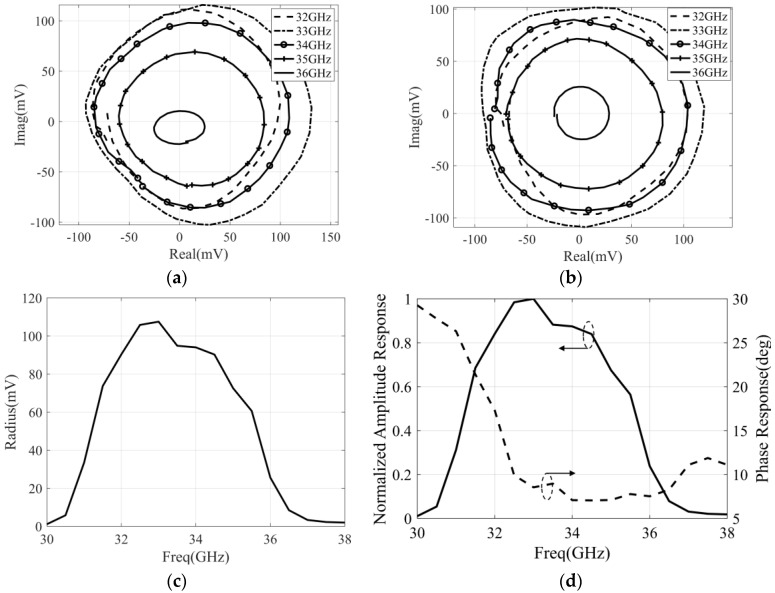
The measurement and calibration results of the correlator under the condition that the power of the input signals is −35 dBm and the frequency varies from 30 to 38 GHz: (**a**) measured correlation circle; (**b**) calibrated correlation circle; (**c**) the variation in correlation circle radius with frequency; (**d**) the amplitude response and phase response versus frequency.

**Figure 15 micromachines-14-00867-f015:**
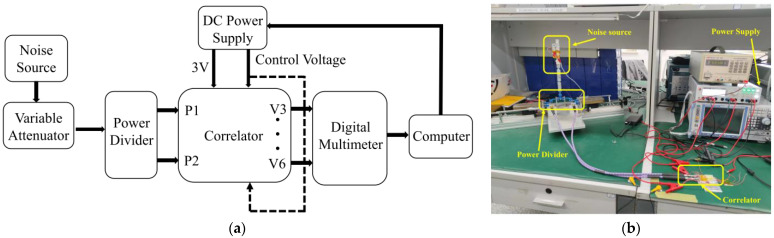
The proposed broadband noise measurement system: (**a**) block diagram; (**b**) test site.

**Figure 16 micromachines-14-00867-f016:**
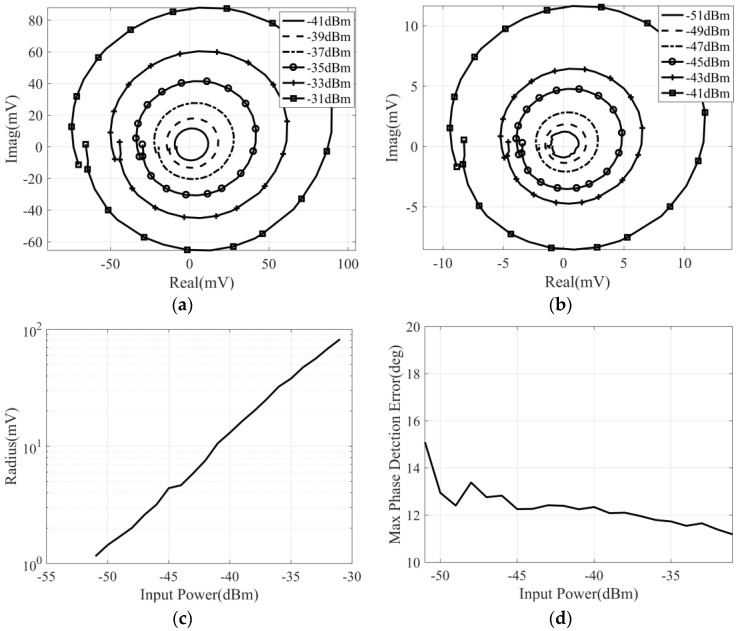
The broadband noise measurement and calibration result of the correlator with input noise power varies from −51 dBm to −31 dBm: (**a**) measured correlation circle with input power varies from −41 dBm to −31 dBm; (**b**) measured correlation circle with input power varies from −51 dBm to −41 dBm; (**c**) the correlation circle radius versus noise power; (**d**) the variation in maximum phase detection error with input power.

**Figure 17 micromachines-14-00867-f017:**
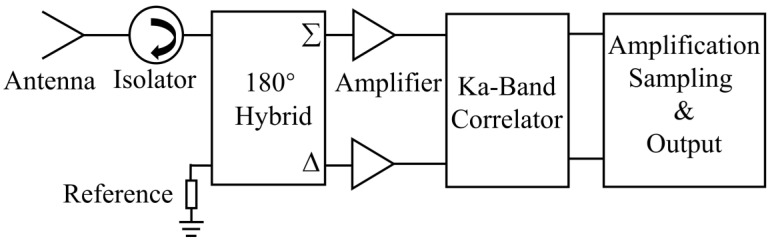
The block diagram of the millimeter-wave radiation temperature measurement system.

**Figure 18 micromachines-14-00867-f018:**
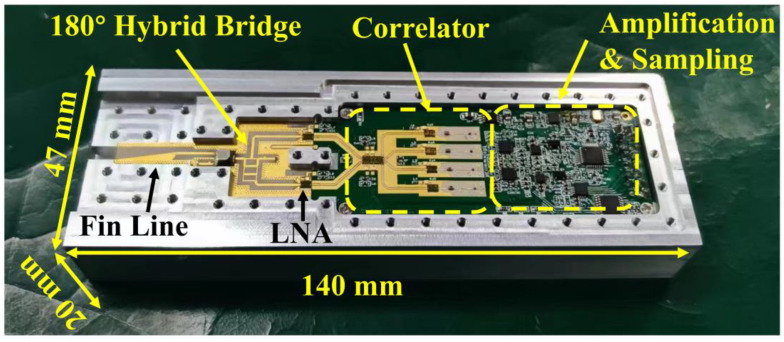
The final designed thermometer.

**Figure 19 micromachines-14-00867-f019:**
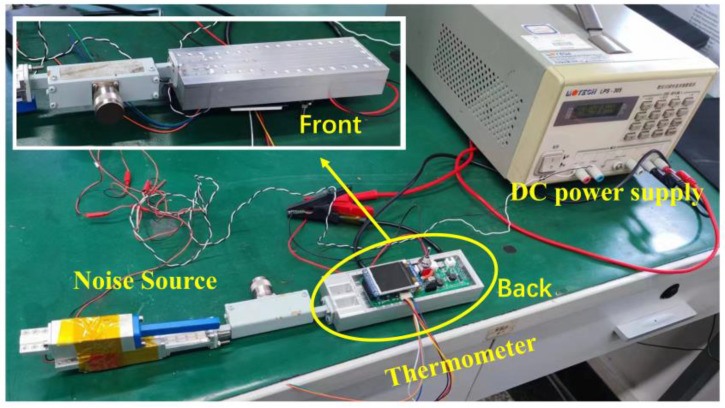
The test site of the thermometer.

**Figure 20 micromachines-14-00867-f020:**
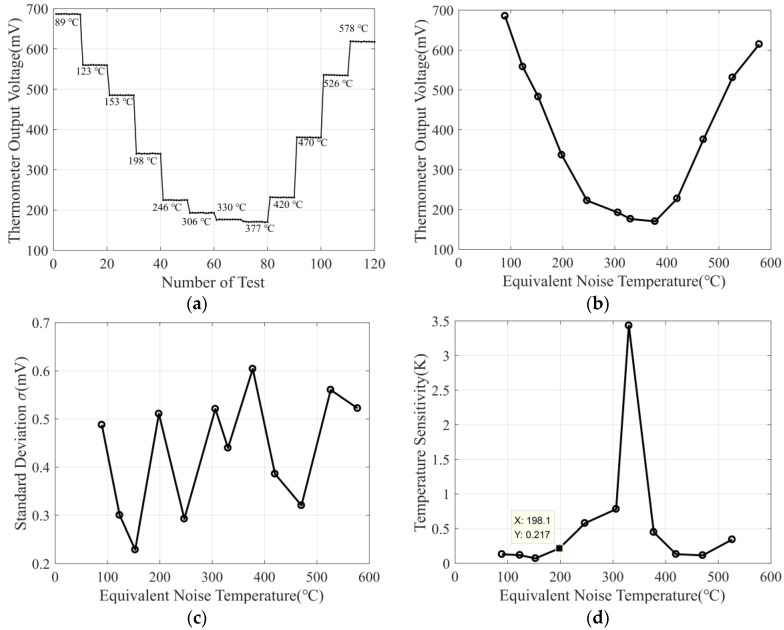
The measurement result of the thermometer: (**a**) thermometer output voltage versus equivalent noise temperature; (**b**) thermometer average output voltage versus equivalent noise temperature; (**c**) standard deviation of thermometer versus equivalent noise temperature; (**d**) thermometer temperature sensitivity versus equivalent noise temperature.

**Table 1 micromachines-14-00867-t001:** Comparison of the analog correlator.

Reference	This Work	[37]	[21]	[13]
Freq(GHz)	32–36	4–8	3.5–8	1.5–2.5
Input Power * (dBm)	−35	−30	−20	−17
Equivalent Bandwidth (GHz)	3.42	3.94	4.2	0.92
Correlation Efficiency	92.5%	99.3%	96.6%	95.9%

* The input power during measurement.

## Data Availability

Not applicable.

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
