# Peer review of "Design of a Compact Analog Complex Correlator for Millimeter-Wave Radiation Temperature Measurement System"

_micromachines, 2023, doi:10.3390/mi14040867_

Round 1
Reviewer 1 Report
· Title is not justified
· Abstract is more like a conclusion, authors should re-write the abstract
· Only few literature study can be a backdrop of current study, the rest are all general ref’s
· Equations 1-9 are used in which program or tool to simulate the parameters? Also mention the source of equations
· Figure.2 is not clear
· Fig.1 and 5 are repeated
· Section 2 includes correlator and section 3 includes design of ka band which makes confusion, author should clear identify or merge the two sections
· Fig.11 a & b, variation starts from 40-50dBm then how the pattern is different?
· The same scenario in figure.12,13,14,15 a & b
· Mention other applications (except the one listed) of this design study
Reviewer 2 Report
The paper presents an analog complex correlator based on the integrated six-port chip is presented for Ka-band temperature measurement system.
Here are my comments and concerns:
1. The paper lacks originality and does not add to the existing body of knowledge.
2. Most of content, theory section, formulas and picture are the same as previous publications of authors (ref [27] and [30]).
3. I suggest to improve the presentation of the paper, such as improving the clarity of the writing or providing more detailed explanations of the equations and methodology.
4. What is the benefit of measuring the body temperature in less than 0.2 K? Is there any applications that needs this sensitivity? the authors could elaborate on the importance of accurate and reliable human body temperature measurement, and how the use of an analog complex correlator can improve upon existing methods.
5. It is mentioned the dynamic range of input power of the correlator is -70 dBm 14 to -35 dBm. What is the limitation, especially for high power inputs?
6. What is the benefit of a few giga hertz bandwidth for this application? Does measuring the temperature need the high data rate or BW?
7. Please cite more relevant research papers, for example, the following related papers:
[1] M. Akbari et al., "Highly Efficient Front End Direct Conversion Receiver for 28-GHz Wireless Access Point," in IEEE Access, vol. 9, pp. 88879-88893, 2021, doi: 10.1109/ACCESS.2021.3089582.
[2] P. S. Chew, K. S. Yeo, K. Ma and Z. H. Kong, "A 57 to 66 GHz novel six-port correlator," 2015 IEEE 11th International Conference on ASIC (ASICON), Chengdu, China, 2015, pp. 1-4, doi: 10.1109/ASICON.2015.7516890.
[3] M. D. Ardakani, et al., "Accurate Millimeter-wave Carrier Frequency Offset Measurement Using the Six-port Interferometric Technique," 2018 48th European Microwave Conference (EuMC), Madrid, Spain, 2018, pp. 1061-1064, doi: 10.23919/EuMC.2018.8541722.
[4] J. Moghaddasi and K. Wu, "Multifunction, Multiband, and Multimode Wireless Receivers: A Path Toward the Future," in IEEE Microwave Magazine, vol. 21, no. 12, pp. 104-125, Dec. 2020, doi: 10.1109/MMM.2020.3023223.
[5] H. Arab, et al., "Early-Stage Detection of Melanoma Skin Cancer Using Contactless Millimeter-Wave Sensors," in IEEE Sensors Journal, vol. 20, no. 13, pp. 7310-7317, July 2020, doi: 10.1109/JSEN.2020.2969414.
8. The measured return loss of final detector is not perfect over the band. How do you claim over 92% of efficiency including this mismatch?
9. There is no impact of noise in detected signals at low powers (Fig 12). Can you please clarify the measurement procedure?
10. There is no information on the LNAs, gain blocks and sampling circuits. At baseband and using operational amplifiers, how do you keep the bandwidth fixed? Can you please provide a study over the MDS and SNR of the full circuit?
11. It would be useful to include a brief discussion of the limitations of the research and potential areas for future work.
Round 2
Reviewer 2 Report
All of my concerns and comments have been addressed by the authors. I recommend minor modifications to the grammar and text, as there are some typos in the article.
Regarding Figure 2, I suggest using a white background in ADS and copying directly from the ADS schematic to Word to improve its quality.
Thank you and good luck.